# Unraveling the Role of Antimicrobial Peptides in Insects

**DOI:** 10.3390/ijms24065753

**Published:** 2023-03-17

**Authors:** Sylwia Stączek, Małgorzata Cytryńska, Agnieszka Zdybicka-Barabas

**Affiliations:** Department of Immunobiology, Institute of Biological Sciences, Faculty of Biology and Biotechnology, Maria Curie-Sklodowska University, Akademicka 19 St., 20-033 Lublin, Poland

**Keywords:** antimicrobial peptides, insect immunity, aging, gut homeostasis, antiviral peptides, *Drosophila*, endosymbionts, functions of AMPs, *Galleria mellonella*, neurodegeneration

## Abstract

Antimicrobial peptides (AMPs) are short, mainly positively charged, amphipathic molecules. AMPs are important effectors of the immune response in insects with a broad spectrum of antibacterial, antifungal, and antiparasitic activity. In addition to these well-known roles, AMPs exhibit many other, often unobvious, functions in the host. They support insects in the elimination of viral infections. AMPs participate in the regulation of brain-controlled processes, e.g., sleep and non-associative learning. By influencing neuronal health, communication, and activity, they can affect the functioning of the insect nervous system. Expansion of the AMP repertoire and loss of their specificity is connected with the aging process and lifespan of insects. Moreover, AMPs take part in maintaining gut homeostasis, regulating the number of endosymbionts as well as reducing the number of foreign microbiota. In turn, the presence of AMPs in insect venom prevents the spread of infection in social insects, where the prey may be a source of pathogens.

## 1. Introduction

Antimicrobial peptides (AMPs) are indispensable components of insect innate immunity. They exhibit antibacterial, antifungal, antiviral, and antiparasitic activity. Since the discovery of the first insect AMP, cecropin in *Hyalophora cecropia* hemolymph [1], many AMPs with different biochemical and antimicrobial properties have been described in species representing different insect orders. Several families of AMPs have been identified in *Drosophila melanogaster* (Diptera), e.g., cecropins, defensin, drosocin, drosomycin, diptericins, metchnikowin, and attacins [2]. In lepidopteran species, e.g., the greater wax moth *Galleria mellonella* or the mulberry silkworm *Bombyx mori*, cecropins, defensins, gloverins, moricins, proline-rich peptides, attacins, and anionic antimicrobial peptides have been characterized [3,4,5,6]. The Antimicrobial Peptide Database [7] (https://aps.unmc.edu/; accessed on 11 February 2023) collecting AMPs of natural origin contains 3569 peptides, including 367 insect-derived molecules.

Most AMPs are amphipathic cationic molecules. Amphipathicity is important for the interaction of an AMP with microbial phospholipid membranes, which are regarded as the main targets of many AMPs. Binding with microbial membranes occurs through involvement of electrostatic and hydrophobic interactions. Many modes of AMP interactions with membranes have been described, e.g., (i) carpet model, (ii) barrel-stave model, (iii) thoroidal-pore model, (iv) aggregate model, (v) charged lipid clustering, (vi) membrane thinning/thickening, and (vii) formation of non-bilayer intermediates [6,8,9,10]. The peptide-membrane interaction often disturbs the membrane structure and leads to its depolarization and changes in permeability, eventually causing microbial cell death. Some AMPs can traverse the cell membrane and, once in the cell, they target intracellular components causing disturbance or inhibition of key metabolic processes: replication, transcription, protein biosynthesis and folding, and synthesis of cell wall components. In addition to cationic AMPs, which comprise the majority of natural AMPs characterized so far, anionic AMPs have been described. It is postulated that such peptides may have evolved in response to pathogens that developed resistance mechanisms against cationic AMPs [8,11,12]. Insect AMPs can be divided into three classes: (i) linear peptides without cysteines forming α-helices, (ii) peptides containing cysteines and stabilized by disulfide bridges, e.g., peptides with β-sheet structures or αβ motifs, and (iii) peptides with overrepresentation of one amino acid, usually glycine and/or proline. Despite the broad structural diversity, studies on the structure-activity relationships revealed that (i) net charge, (ii) hydrophobicity, and (iii) amphipathicity are the most important physicochemical and structural determinants of antimicrobial activity and effectiveness as well as selective toxicity of AMPs [6,8,13]. Moreover, using bioinformatic and pattern recognition methods, multidimensional structural signatures were identified in AMPs, which appeared to be the requisites for antimicrobial activity. One of such unifying signatures was found in eukaryotic α-helical AMPs. This unifying α-core signature contains a helical domain of 12 residues with a mean hydrophobic moment of 0.50 and favoring aliphatic over aromatic hydrophobic residues [14]. Similarly, a conserved signature, called the γ-core motif, was identified in AMPs stabilized with disulfide bridges. The γ-core motif is composed of two antiparallel β-sheets with a short turn region. The sequence signatures include (i) the length of 8–16 amino acid residues, and (ii) conserved GXC or CXG motifs within the sequence [15].

In insects, two main signaling pathways, Toll and Immune deficiency (Imd), are involved in the regulation of AMP gene expression in response to infection. The Toll and Imd pathways are activated by Lys-type peptidoglycan of Gram-positive bacteria and DAP-type peptidoglycan of Gram-negative bacteria (and Gram-positive bacteria *Bacillus* and *Clostridium*), respectively, after recognition by proper pattern recognition receptors (PRRs). The Toll pathway also responds to fungal infection after recognition of β-1,3-glucan. In *Drosophila*, Lys-type PG is recognized by extracellular peptidoglycan recognition proteins (PGRPs) PGRP-SA and -SD, whereas DAP-type PG is recognized by transmembrane PGRP-LC and intracellular PGRP-LE. Some PGRPs (PGRP-SB, -SC, -LB, -LD) possess amidase activity, which is essential in gut microbiota control [16]. Toll and Imd activation leads to nuclear translocation of Dorsal/Dif and Relish transcription factors, respectively, i.e., members of the NF-κB family. Induction of AMP gene expression occurs locally in epithelial cells, fat body cells, and hemocytes and results in a systemic response to infection. In epithelia, where AMPs can be expressed constitutively, the Imd pathway is mainly engaged. In response to infection, the Jak-STAT pathway can be activated as well (Figure 1) [17,18,19,20]. However, AMP gene expression can be induced by other signals, not necessarily connected with invading pathogens. In non-infected insects, metabolic changes, stressors, and aging can induce the expression of AMPs. Such a response is regulated by two other signaling pathways, i.e., the insulin-like signaling (ILS) pathway and the target of rapamycin (TOR) pathway, in which transcription factors from the Forkhead Box (Fox) family are involved [21].

The action of AMPs (also insect AMPs) serving as effective antimicrobials against various pathogens has recently been presented in several excellent reviews [6,11,22]. However, there is growing evidence that these molecules can play other physiological roles, being even dangerous for host tissues, e.g., regulation of microbiota, involvement in nervous system activity, and aging. Most of the evidence on these sometimes unexpected functions of AMPs has been obtained in research on *D. melanogaster*, since there are numerous different mutants and genetic tools available facilitating the development of Drosophila models of human diseases. However, there is some evidence from studies on other insect species, e.g., lepidopterans *G. mellonella* and *B. mori* or hymenopteran *Apis mellifera*. In this review, we will focus especially on other functions of AMPs in insects discovered to date (Figure 2).

## 2. AMPs in the Nervous System

Research conducted so far shows that there is a connection between alterations in immune signaling and neuronal health, communication, and activity in insects. Changes in neuroimmune communication and glial cell signaling may contribute to behavioral changes in the insect. In addition, the insect nervous system can mount a local immune response to infection, and activation of signaling pathways may lead to neurodegeneration, malfunctioning neurons, and altered behavior [23]. Moreover, the AMP expression pattern is affected by aging independently of infection, and it has been postulated that an increased level of some AMPs produced in non-neuronal tissues during aging can mediate a signal initiating neuronal aging [24].

AMPs are important for regulation of normal functions of the insect brain, e.g., sleep and non-associative learning in *Drosophila* [25,26]. Results reported by Dissel et al. showed that the transcript levels of *metchnikowin* (*Mtk*), *drosocin* (*Dro*), and *attacin* (*Att*) were differentially increased in glia, neurons, and the head fat body, respectively, in sleep-deprived flies. They further demonstrated that the expression of *Mtk* in glia but not in neurons and the expression of *Dro* in neurons but not in glia had a negative effect on memory but modulated sleep in an opposite way. These AMPs were considered as candidates for conferring resilience/vulnerability to sleep deprivation [27]. A further link between sleep regulation and immune response in *Drosophila* was found by Toda et al. in their research on sleep mutants and the role of *nemuri*–a gene involved in sleep induction. *Nemuri* overexpression increased the sleep length and depth in *Drosophila*. Interestingly, *nemuri* encodes an AMP that can be secreted ectopically to drive prolonged sleep and to promote survival after infection. This peptide acts non-cell-autonomously to promote sleep. The nemuri peptide contains 64 amino acids, has sequence similarity to a Greenland cod (*Gadus ogac*) cathelicidin, and kills bacteria in vitro (*Serratia marcescens*, *Escherichia coli*) and in vivo. *Drosophila* adults overexpressing *nemuri* in neurons and infected with *S. marcescens* or *Streptococcus pneumoniae* survived longer, had a longer amount of sleep, and contained a lower bacterial load than control insects [28]. In another study, a *Drosophila* non-associative long-term memory (LTM) paradigm involving a natural predator of *Drosophila*, i.e., the endoparasitoid wasp *Leptopilina heterotoma*, was used for identification of novel memory genes during and after memory formation. The examination of transcriptional changes in the fly brain revealed that *cecropin A1* (*CecA1*), *CecA2*, *Dro*, *AttA*, *dipericin* (*Dpt*), and *DptB* were differentially regulated at various time points. Specifically, *CecA1*, *Dpt*, and *Dro* were downregulated and upregulated in 7-h and 4-day exposed individuals, respectively [29]. Evidence that AMPs can directly affect brain function was also presented by Barajas-Azpeleta et al., who found 10-12-fold upregulation of *diptericin B* expression in the adult fly head following behavioral training. Moreover, using *DptB* null flies, they demonstrated that *DptB* was required for modulation of long-term memory in *Drosophila*. Interestingly, only silencing the *DptB* expression in the head fat body affected long-lasting memory, clearly indicating the head fat body as the only relevant source of *DptB* for behavioral modification [30].

Recently, the expression of two AMPs belonging to defensins, i.e., galiomicin and gallerimycin, has been detected in the brains of *G. mellonella* larvae after injection of *Habrobracon hebetor* venom or topical application of the entomopathogenic nematode *Steinernema carpocapse*. However, the role of these peptides in the *G. mellonella* nervous system has not been investigated yet [31]. In this study, the upregulation of the *sericotropin* gene and peptide was also detected in the brains of challenged larvae. Sericotropin functions in lepidopteran insects as a stimulator of silk production. It has been demonstrated that synthetic peptides derived from the N- and C-terminal parts of sericotropin exhibited antibacterial activity against *Xenorhabdus* spp. bacteria, i.e., symbionts of *S. carpocapse*, suggesting an antimicrobial role of sericotropin expressed in the *G. mellonella* brain [31]. In the brains of the honeybee *A. meliffera* infected with deformed wing virus (DWV), transcriptomic and metabolomic analyses revealed overexpression of genes encoding proline-rich AMPs, abaecin and apidaecin, as well as hymenoptaecin and an increased level of these peptides [32]. Results obtained in a study on the response of the central nervous system (CNS) of *Locusta migratoria manilensis* infected with the fungus *Metarhizium acridum* also indicated an important cross-talk between the CNS and the immune system. The expression patterns in the CNS responded rapidly to the infection and changed as the infection proceeded. Many differentially expressed genes, directly involved in immune function and regulation, were identified, including genes encoding defensin and diptericin, which were up-regulated in the CNS of *M. acridum*-challenged locusts [33].

The use of *Drosophila* models of Alzheimer’s disease (AD), traumatic brain injury (TBI), and ataxia-telangiectasia (AT) revealed a role of the Toll and Imd signaling pathways in neurodegeneration [34,35,36,37,38]. Although Toll signaling is engaged in proper development of the brain in *Drosophila*, it was demonstrated that suppression of Toll signaling protected against neurodegeneration. Moreover, *Drosophila* mutants in the negative regulator of the Imd pathway, *dnr1* (defense repressor 1), had increased expression of AMP genes (*Cec*, *Dpt*, *Att*, *drosomycin* (*Drs*), *Mtk*), which was correlated with progressive age-dependent neuropathological changes and a shortened lifespan. It has also been demonstrated that neurodegeneration is dependent on the transcription factor Relish, and bacterial infection in the brain can trigger neurodegeneration through the neurotoxic effects of AMPs. On the other hand, *Relish* mutations inhibited upregulation of innate immune response genes and neurodegeneration in AT *Drosophila* mutants, whereas overexpression of active Relish in glial cells resulted in neurodegeneration [35,39]. Similar effects as those caused by mutations in *dnr1* were observed in flies with mutations in other negative regulators of the Imd signaling pathway: *zfh1* (transcription factor Zn finger homeodomain 1), *Pirk* (poor Imd response upon knock-in), *trbd* (deubiquitinase Trabid), and *tg* (Transglutaminase), strengthening the evidence for the negative role of AMPs in the development of diseases [40,41]. Moreover, the overexpression of *Dro*, *AttC*, and *CecA1* in neurons or glia was accompanied not only by a shortened lifespan but also by early appearance of locomotor defects in comparison with controls, suggesting a causative relationship between high levels of AMPs and neurodegeneration in *Drosophila* [41]. In turn, Barati et al. studied the effect of amyloid-β 42 (Aβ42) and tau on the Imd pathway and neuroinflammation gene expression in a *Drosophila* model. They showed that the expression of genes involved in the Imd pathway, such as *Relish* and AMPs (*AttA*, *DptB*), increased with age in the W^1118^ control flies and in the embryonic nervous system of AD transgenic flies Aβ42, tau^WT^, or tau^R406W^, but the level of AMPs in glia remarkably decreased compared to W^1118^. The decline was higher in both tau flies compared to Aβ42 transgenic flies. The overexpression of AMPs in *Drosophila* leads to brain neurodegeneration and neuroinflammation [42]. Another example connecting overexpression of AMPs with neurodegeneration was provided by research on the role of the Yorkie transcription factor in polyglutamine (PolyQ)-mediated neurotoxicity (involved in Huntington’s disease in humans) in *Drosophila*. It was found that PolyQ expansion increased the expression of *CecA*, *Att*, *Dpt*, *Dro*, and *Drs*. It was postulated that upregulation of AMPs can participate in PolyQ-mediated neurotoxicity in *Drosophila* [43].

An important role of one of the AMPs, metchnikowin, in acute and chronic outcomes of TBI was demonstrated by Swanson et al. in a *Drosophila* TBI model. In an analysis of null mutations in 10 AMP genes (*AttA*, *AttB*, *AttC*, *AttD*, *defensin* (*Def*), *Dro*, *Drs*, *Mtk*, *DptA*, *DptB*), it was found that *Mtk* mutant flies exhibited reduced acute behavioral deficits and only the mutation in the *Mtk* gene protected flies from early mortality after TBI in different diet conditions. These results indicate that *Mtk* plays an infection-independent role in the nervous system and can promote neuropathological changes in the brain following TBI [44]. In another study conducted in a *Drosophila* Closed Head Injury (dCHI) model, an acute broad-spectrum immune response was detected in glia cells, where many AMP genes were up-regulated 24 h after TBI (*Att*, *Cec*, *Def*, *Dpt*, *Dro*, *Drs*, *Mtk*, *listericin* (*Lys*)). It was found that the deletions of most of these AMPs shortened insect survival after TBI. In contrast, loss of *Def* extended survival, indicating that the *Drosophila* immune response to TBI, including the induction of AMP expression, may result in different effects. Moreover, in this research, a link between the immune response and sleep in flies was demonstrated, as loss of *Relish* protected against impaired control of sleep and movement after TBI [26].

Progress of neurodegenerative processes is correlated in time with an age-dependent increase in AMP expression in the head tissue of *Drosophila* [41]. Using a transgenic *Drosophila* model of AD, Wang et al. provided evidence for distinct expression profiles of AMP genes in an aging model and in wild-type flies. Whole transcriptome profiles of wild-type *Drosophila* heads revealed upregulation of 12 AMP genes: *AttA*, *AttB*, *AttC*, *AttD*, *CecA2*, *CecC*, *DptA*, *DptB*, *Drs*, *Def*, *Dro*, and *Mtk*. A gradual increase in expression of these AMPs was observed in the wild-type flies during the normal aging process, whereas an initial decrease and a subsequent increase in the expression levels was found during aging of the AD flies. Interestingly, significant correlations between the abnormal amyloid β (Aβ) peptide concentration and abnormal expression of *AttB*, *AttC*, *CecA*, *Drs*, *Mtk*, and *LysS* were detected, suggesting contribution of AMP dysregulation to AD progression by inducing deposition of Aβ peptide aggregates [45]. However, in other studies, lysozyme was demonstrated to be beneficial in different *Drosophila* models of AD. For example, co-expression of human lysozyme rescued the rough eye phenotype indicative of toxicity in flies that expressed Aβ_1-42_ in the eyes. Moreover, lysozyme binding with Aβ_1-42_ in the *Drosophila* eye was detected, suggesting that such an interaction prevented from the toxic effects of Aβ_1-42_ [46]. Interestingly, antimicrobial activity of the Aβ peptide was reported, suggesting that it is an AMP in the nervous system and that neurodegenerative Alzheimer’s disease may be partially an infectious disease [47]. Increased activation of innate immunity mechanisms reflected by overexpression of AMP genes was also postulated as a cause of long-term deleterious effects leading to neurological deficits, persistent cell death in brains, and premature aging in a *Drosophila* model of radiation-induced neurotoxicity. In this study, upregulation of *Drs*, *DroA*, *Dpt*, *AttC*, *Cec*, and *Mtk* was detected in brains of 5-day- and 15-day-old adult flies after radiation exposure during larval development (late third larval instar), indicating contribution of persistent activation of immune signaling pathways to neurological deficits in adult flies [48].

The connection between immunity and nervous system is also evident at the level of neuropeptides. Among human neuropeptides, antimicrobial activity has been reported for enkelytin, substance P, neuropeptide Y, bombesin, adrenomedullin, vasoactive intestinal peptide, and pituitary adenylate cyclase activating polypeptide (PACAP) [49,50]. Besides regulation of neurodevelopment, emotion, and certain stress responses, the PACAP neuropeptide acts as an antimicrobial agent. Although its antimicrobial function was demonstrated in mammals, the phylogenetic analysis predicted antimicrobial properties of eleven PACAP homologs, including PACAP from the cockroach *Periplaneta americana* [51].

## 3. AMPs in Aging

Aging is a complex process involving accumulation of harmful changes resulting in an overall decline in several vital functions. It leads to progressive deterioration in the physiological condition of the organism such as changes in metabolism and behavior, impaired neuronal function, and reduced stress resistance, reproductive and immune capacity, or barrier function in the gut, which eventually cause disease and death. Many results important for understanding the molecular basis of aging were provided by research conducted in a *D. melanogaster* model [52].

A well-documented hallmark of aging in *Drosophila* is the decline in immune functions accompanied by an increase in AMP gene expression. Aging is associated with expansion of the AMP repertoire and loss of their specificity, possibly caused by immune dysregulation [2,53,54]. It was reported that the expression level of AMP genes, including *AttA*, *AttB*, *AttC*, *AttD*, *CecC*, *Dpt*, *Def*, *Dro* and *Met* as well as transcription factor *Relish*, increased considerably with age in *Drosophila* flies [55,56]. Initially, elevated AMP levels in old flies were associated with an impaired ability to clear infections effectively, but later studies showed little correlation between the AMP expression level and the microbial load. Badinloo et al. showed that decreased expression of *Relish* in the fat body of old *Drosophila* flies extended longevity, which was correlated with lower activation of AMP genes, whereas overexpression of AMP genes controlled by Relish was accompanied with a significantly shortened lifespan. Particularly, the overexpression of four AMPs: attacin A, defensin, metchnikowin, and cecropinA1, significantly decreased the lifespan of the flies, while the overexpression of drosomycin and drosocin had no significant impact on longevity. Increased expression of AMPs was observed in different body parts, including the gut, muscles, and fat body. High AMP levels correlated with depolarization of mitochondrial membranes and apoptosis of cells in different tissues. Hence, AMPs may have an impact on aging by causing changes in the mitochondrial membranes that lead to apoptosis [57]. The death of olfactory receptor neurons, particularly Or42b, was demonstrated to be connected with increased expression of AMPs during non-pathological aging in *Drosophila*. It was found that these neurons died as a result of age-related caspase 3-like protease activation associated with systemic activation of innate immunity signaling pathways, leading to enhanced expression of AMPs. Interestingly, *Dro* was sufficient for activation of caspase 3-like protease and cell death in Or42b neurons [58]. Overexpression of AMP genes was also connected with a shortened lifespan when the circadian clock-controlled *Achilles* gene was characterized in *Drosophila*. *Achilles* encodes a putative RNA-binding protein; its activity is rhythmic at the mRNA level, and it suppresses the expression of AMPs. Knocking-down of *Achilles* in *Drosophila* neurons led to dramatically increased systemic expression of immune related genes, including *Mtk* and *Drs*, and significantly reduced the overall lifespan of adult flies in the absence of immune challenge [59], again indicating a role for AMPs in lifespan shortening.

On the other hand, activation of a single AMP, *Dro*, in the gut or ubiquitously in adult *Drosophila* caused a significant extension of the fly lifespan accompanied by lower activity of immune signaling pathways over lifetime, decreased stress response, and slower loss of gut barrier integrity. No such effect was observed when *Dro* was expressed in the gut of flies with a diminished endogenous bacterial load in the gut, indicating a cross-talk between innate immunity, intestinal homeostasis, and aging. Similarly, increased ubiquitous expression of *CecA1* positively influenced the *Drosophila* lifespan [60]. A role of AMPs, mainly *Dpt*, in increasing tolerance to oxidative stress correlated with a prolonged lifespan of *Drosophila* adults was reported in earlier studies. Overexpression of *Dpt*, *Att*, *Cec*, *Drs*, and *Mtk* was detected in flies that tolerated enormous oxidative stress. Then, it was demonstrated that overexpression of even one AMP (*Dpt*, *Att*) can be sufficient for better survival of wild-type flies in hyperoxia [61,62].

Recently, using *Drosophila* lines with different combinations of knocked out AMP genes, the pathogen-specific roles of AMPs have been reported, where microbicidal activity against a particular pathogen depended mainly on a certain AMP, e.g., the *Dpt* gene family alone provided effective defense against *Providencia rettgeri*, drosocin contributed to defense against *Enterobacter cloacae*, whereas buletin (also encoded by the *Dro* gene) was effective against *Providencia burhodogranariea* [63,64]. In addition to these findings, it turned out that when considering the role of AMPs in *Drosophila* response to infection, age and sexual dimorphism should be taken into account given the documented strong differences in immune response at the basal level upon infection and during aging between sexes [65,66]. Hanson et al. showed that *Dpt* expression was sufficient to protect young male flies against *P. rettgeri*; however, Shit et al. demonstrated that reintroduction of functional *Dpt*A and *Dpt*B (∆*AMPs^+Dpt^*) fully restored survival and decreased the *P. rettgeri* load only in young males but not in females or older males [63,66]. Aging had negative effects on the fitness and pathogen clearance ability of *Drosophila* females but not males. Infection of flies with *P. rettgeri* resulted in higher mortality and increased bacterial burden in older females than in younger females, while both young and old males had similar mortality rates and bacterial loads. Most mutant flies lacking a certain AMP, or a combination of AMPs, became highly susceptible to infection with age, with the exception of female mutants lacking *Def.* This effect may be related to the sexual dimorphism, where defensin is important for defense in males but not in females [66].

As reported by Badinloo et al., AMP induction in old flies appeared to be independent of recognition of PAMPs upstream Imd, but the activity of Ird5, i.e., a component of the IKK complex, was essential for AMP expression. The up-regulation of AMPs associated with aging was reduced in the *ird5* mutant but was unaffected in the *imd* mutant flies, indicating that aging activates other pathways involved in induction of AMP expression [57]. Much earlier, Becker et al. showed that the AMP expression in non-infected *Drosophila* flies may be independent of the Toll and Imd signaling pathways but may depend on the transcription factor Forkhead Box O (FoxO), which is known as a key downstream regulator of the ILS pathway involved in metabolism, stress resistance, and aging. The AMP upregulation was lost in starved *foxo* mutant larvae, while *foxo* overexpression led to increased AMP expression. Direct binding of FoxO to the *Drs* regulatory region in *Drosophila* cells suggested direct regulation of AMP genes by FoxO, depending on the energy status of the cell, and indicated a cross-regulation of metabolism and innate immunity [67]. Conserved FoxO-binding sites have been identified in silico in *Drosophila* promoter regions of *Cec* and *Dro* [68]. These authors also demonstrated that AMPs were produced in *Drosophila* enterocytes in a FoxO- but not Relish-dependent manner. Furthermore, they showed the FoxO regulatory role in gut AMP expression after *Drosophila* oral infection with the Gram-negative bacterium *S. marcescens*. After infection, upregulation of *Dpt*, *AttA*, and *AttB* was detected in gut epithelia of wild type flies, while no significant increase in the transcript levels of these genes was noticed in the *foxo* mutants. The impaired FoxO signaling diminished the resistance of *Drosophila* to intestinal infection due to an insufficiently high level of AMP expression, which eventually caused a decline in survival [68]. Recently, involvement of FoxO in the activation of AMP gene expression has been demonstrated in the fat body of starved *B. mori* larvae [69]. Similarly to FoxO, a role of Forkhead (FKH), i.e., another transcription factor from the same family, in the regulation of AMP expression in the *Drosophila* gut was demonstrated. The results indicated that the TOR signaling pathway, responding mainly to protein availability, is also engaged in the cross-talk between metabolism and innate immunity by regulation of the expression of *Dpt* and *Mtk* [70].

## 4. AMPs as Antitumor Agents

Some insect AMPs exhibit selective in vitro cytotoxicity against different cancer cell lines, e.g., melanoma, lymphoma, leukemia, breast cancer, lung cancer, and bladder cancer [71,72,73,74]. Besides in vitro studies on the anticancer activity of insect AMPs, there are reports indicating in vivo involvement of AMPs in antitumor action, particularly based on research carried out using *Drosophila* models.

In vivo studies conducted by Parvy et al. showed the anticancer effect of defensin in a *D. melanogaster* model. The presence of tumor cells in the *D. melanogaster* body activates a cellular response, which leads to release of Eiger, i.e., a Tumor Necrosis Factor (TNF), from hemocytes. TNF is required for the exposure of phosphatidylserine on the surface of tumor cells. The presence of this phospholipid on the surface of cells gives them a negative charge, making them the target of cationic defensin produced, among others, in fat body cells. The effect of this interaction is cancer cell death and tumor regression. The Imd pathway in the trachea is responsible for expression of the *Def* gene in *D. melanogaster* and the Toll and Imd pathways in the fat body. Deletion of the defensin-encoding gene resulted in death of fewer cancer cells and enlargement of the tumor [75]. It was also found that AMP genes *Drs*, *Def*, *Dpt, Mtk*, *AttA*, and *CecA2* were upregulated in Drosophila *mxc^mbn1^* mutants with malignant hyperplasia in lymph glands. The downregulation of the Toll or Imd pathways increased, while the ectopic expression of each of the five different AMPs (*Drs*, *Def*, *Dpt*, *Mtk*, *AttA*) in the fat body suppressed the tumor phenotype in the mxc mutants. Moreover, drosomycin and defensin secreted by the fat body were taken up by circulating hemocytes and accumulated in their cytoplasm. These hemocytes were recruited into the tumor. Another AMP, diptericin, was directly localized at the tumor region. The AMPs increased apoptosis only in the tumor cells [76,77]. Literature data also indicate a link between anti-infection and anti-cancer mechanisms. Jacqueline et al. reported that oral infection with the Gram-negative bacterium *Pectobacterium carotovorum* accompanying tumor in *Drosophila* resulted in a reduction in the tumor size, compared to insects without infection. This effect was related to an increase in the expression of genes encoding diptericin and drosomycin AMPs, leading to increased death of tumor cells [78].

## 5. AMPs and Symbiotic Microbiota

Many insects possess mutualistic symbiotic microorganisms in the gut lumen and the body cavity or inside cells (Table 1). Most symbionts are vertically transmitted from mother to offspring, but gut symbiotic bacteria can be horizontally acquired in every generation from the environment, such as soil, water, and food. Some symbionts are obligate because they are essential for host survival, while others are facultative. All insects harbor facultative bacterial endosymbionts. Usually, these symbionts are beneficial to the host, providing essential nutrients, defending against pathogens, and facilitating adaptation to environmental conditions. They supplement the insect host’s diet with vitamins and cofactors and are a source of important amino acids [79]. For long-term survival in the host body, symbionts use different strategies that allow them to withstand the adverse effects of the host immune defense mechanisms and simultaneously be safe for the host [80,81,82,83,84]. Nevertheless, symbionts can become harmful to the host if they reproduce and grow uncontrollably, as they consume more nutrients than in a normal symbiotic relationship. Therefore, their number in the insect body should be controlled by the host immune system and a disturbance in their load can lead to dysbiosis. In some insect species, endosymbionts are housed within specialized host cells named bacteriocytes, which make up an organ called the bacteriome located around the host midgut and germ cells, from which they are transmitted to their progeny. Compartmentalization of symbionts in bacteriomes is considered a strategy used by the host to control beneficial symbionts in a limited space, whereas host immune mechanisms acting in other host tissues prevent pathogenic infection [2,85,86,87,88]. In insect-bacterial symbiosis, specific factors are produced by the host immune system to modulate the population of symbionts. AMPs, participating not only in the systemic but also local epithelial immune response are such factors [2,89,90]. In the insect gut, the Imd pathway together with the Jak-STAT pathway and the Duox-ROS system play a key role in the fight against pathogens and simultaneous control of symbiotic bacteria. Therefore, dysregulation of these pathways exerts a negative or positive influence on the gut microbial growth [89,91,92,93,94,95].

It has been reported that symbiotic bacteria do not induce the synthesis of intestinal AMPs, although they activate Relish. The low basal level of AMPs is associated with negative regulation of Relish by Caudal (Cad) (Figure 1). Caudal has been identified as a homeobox transcription factor essential for the formation of the anteroposterior body axis in *Drosophila* embryos, while in adult flies it is predominantly expressed in the posterior midgut, where it is involved in the maintenance of a low level of midgut-specific AMP gene transcription. The antagonistic relationship between Cad and Relish providing balance between immune defense and microbiota homeostasis in the insect gastrointestinal tract has been reported in *D. melanogaster*, *G. mellonella*, and *Anopheles gambiae* [85,96,97,98]. In *Drosophila*, Caudal is responsible for the constitutive low local expression of *cecropin* and *diptericin* in a tissue-specific manner. In flies lacking Caudal, the gut microbiota continuously activates the Imd pathway inducing AMP overexpression, which results in increased apoptosis of gut epithelial cells and a shorter lifespan [96]. RNAi-mediated silencing of *Cad* in *A. gambiae* resulted in a decreased prevalence and altered the species composition of midgut microbiota due to the increased expression of Relish2-controlled AMPs: cecropin-1, cecropin-3, gambicin, and defensin-1. In addition, after Gram-positive *Staphylococcus aureus* bacterial infection, an increased level of defensin-1 and gambicin was observed, which contributed to better survival of mosquitoes [97].

*Enterococcus mundtii* (syn. *Streptococcus faecalis* Andrewes and Horder) is considered a dominant bacterium in the midgut of *G. mellonella* larvae; it is transmitted vertically−from mother to offspring [99,100,101]. Krams et al. showed that the quality and quantity of the diet affected the number of *Enterococci* and had a significant effect on the expression level of AMP genes in the midgut of *G. mellonella* larvae. An elevated level of *CecD*, *galiomicin*, *gallerimycin*, *gloverin*, and *6-tox* expression was correlated with an increased number of *Enterococcus* symbionts and diet diversity. It was postulated that an increased basal expression level of genes encoding galiomicin and particularly gloverin, i.e., an AMP with activity against Gram-positive bacteria such as *Enterococcus*, may be a mechanism that controls the symbionts and acts prophylactically against opportunistic pathogens in *G*. *mellonella* [101]. It has been shown that the Imd pathway also plays an important role in *G. mellonella* gut immunity, because the expression level of Imd pathway genes (*Imd*, *Relish*), in contrast to those related to the Toll pathway (*Spätzle*, *Dif/Dorsal*), was relatively higher in the gut (Figure 1) [98]. Moreover, it was observed that silencing of *Relish* resulted in downregulation of the gut expression of *CecD*, *gloverin*, *gallerimycin*, *transferin*, and *galiomicin* and bacterial overpopulation in the *G. mellonella* gut. Peroral treatment of the larvae with antibiotics eliminated the gut microbiota and significantly lowered the expression of Imd pathway genes, including all AMP genes. In such conditions, the *Caudal* expression in the gut was decreased as well, probably due to the low activity of the Imd pathway and the low level of AMPs [98,101].

The gut microbiota of the honeybee *A. mellifera* is a highly specialized bacterial community composed of e.g., *Snodgrassella alvi*, *Gilliamella apicola*, *Frischella perrara*, *Bifidobacterium* spp., and *Lactobacillus* spp. (Table 1). *Frischella perrara* is the first to colonize the gut immediately after adult emergence, is harbored by old forager bees, and strongly stimulates the host immune system in the presence of other gut bacterial species. A transcriptome analysis showed that *F. perrara* induced the host gene expression more potently than *S. alvi*, in particular genes known to be activated in response to bacteria, including AMP genes encoding apidaecin type 14 (*Apid1*), apidaecin type 73 (*Apid73*), abaecin (*abaecin*), and defensin 1 (*Def1*) [102]. Another study showed that the microbiota present in the gut of honeybees induced the expression of *Apid* and *hymenoptaecin* genes. A significant increase in the expression of these two AMPs was observed in the gut tissue of bees fed the gut symbiont *S. alvi* or normal gut microbiota, compared to bees lacking the gut microflora. No such differences were detected for the expression of *abaecin* and *Def*. However, bees lacking their normal gut microbiota have lower basal expression of AMPs, which may negatively affect their ability to fight infection. Interestingly, resident gut bacteria are relatively resistant to bee AMPs, compared to foreign species [103]. Additionally, different levels of resistance of *A. mellifera* microbiota species to different AMPs were found. In general, *A. mellifera* microbiota was more resistant to apidaecin Ia than Ib, whereas *S. alvi* was more resistant to apidaecin Ib than *G. apicola*. Gram-positive bacteria (*Lactabacillus*, *Bifidobacterium*) were highly resistant to both apidaecin and hymenoptaecin, while Gram-negative species, particularly *S. alvi*, were more sensitive to hymenoptaecin [103].

Referring to the intestinal microbiota of the honeybee, it is also worth paying attention to peptides with antimicrobial activity present in royal jelly. This is important because royal jelly, secreted by the pharyngeal glands of worker bees, is the food for larvae on the first three days of their development and the queen throughout its life; hence, its antimicrobial action is crucial. Several peptides with antimicrobial activity have been identified in royal jelly, e.g., royalisin and jelleines [104,105]. *A. mellifera* royalisin has broad antifungal and antibacterial properties, for instance against *Paenibacillus larvae*, i.e., an inducer of microbial epidemic in honeybee larvae. Royalisin is a typical cysteine-rich defensin-like peptide composed of 51 amino acids and stabilized by three disulfide bridges. This peptide shows high homology to *Sarcophaga peregrina* sapecin and *Protophormia terraenovae* phormicins, which are peptides with antibacterial activity [106]. Most jelleines (I-IV) are derived from the C-terminus of Major Royal Jelly Protein-1 (MRJP-1). They are short peptides composed of 8–9 amino acids. Jelleines, particularly jelleines I and II, have antibacterial and antifungal activity [104,107,108,109].

Modulation of the immune response to maintain endosymbionts was reported in carpenter ants of the genus *Camponotus* (Hymenoptera). These ants host their primary endosymbiont, a Gram-negative bacterium *Blochmannia*, which resides free in the cytoplasm of bacteriocytes in the midgut tissue and in matured oocytes. This endosymbiont is crucial for ant colonies, ensuring proper growth, development, and high fecundity of its host, which was demonstrated in studies of ant colonies with experimentally-reduced *Blochmannia* [110,111]. *Camptonotus floridanus* ants reduce the immune response within midgut tissue and ovaries in order to allow survival of *Blochmannia floridanus*, while the immune gene expression in other tissues is maintained at a normal level (Table 1) [112]. Low expression of genes encoding lysozymes and such AMPs as hymenopteacin and defensin-1 in bacteriocytes was demonstrated. *C. floridanus* hymenopteacin has activity against Gram-negative bacteria; therefore, its excessive synthesis can harm the Gram-negative endosymbionts. In addition, since the expression level of two PRR genes, *PGRP-LB* and *PGRP-SC2*, in the midguts of untreated insects was high, it was suggested that the products of these genes down-modulate the immune response of this tissue, which ensures tolerance of endosymbionts. These PGRPs are known to possess amidase activity responsible for peptidoglycan degradation [16], which prevents the activation of the immune response towards endosymbionts, including AMP synthesis in *C. floridanus*. Both these genes were much more strongly expressed in the midgut tissue than in other tissues. However, when *Blochmannia* is recognized in the hemocoel, an immune response is activated to prevent the spread of symbionts [112]. More recently, compartmentalized PGRP expression has been detected in the gut of the oriental fruit fly *Bactrocera dorsalis* (Diptera). High expression of *PGRP-LC* was found in the *B. dorsalis* foregut, while *PGRP-LB* and *PGRP-SB*, i.e., encoding receptors with amidase activity, were mainly expressed in the anterior and middle midgut. This expression pattern correlated with the presence of symbiotic Enterobacteriaceae in the *B. dorsalis* midgut. The knockdown of *PGRP-LB* and *PGRP-SB* led to increased expression of midgut AMPs and, consequently, to a decreased number of symbiotic bacteria and an elevated number of opportunistic microbes. Such regional expression of different PGRPs provides protection for symbiotic bacteria [113].

The Gram-negative bacterium *Sodalis pierantonius* is an endosymbiont of the cereal weevils *Sitophilus* spp. (Coleoptera) maintained in the bacteriome. When *S. pierantonius* bacteria are injected into the hemocoel, they induce systemic expression of AMP genes. Nevertheless, in physiological conditions, only the *coleoptericin A (ColA)* AMP gene is expressed in the bacteriome, while the other AMP genes are expressed only slightly or not at all, allowing the endosymbiont to survive. In symbiotic insects, *ColA* was expressed in all tissues harboring the endosymbionts, while in aposymbiotic insects ColA is constitutively present in gut epithelia, the fat body, and under the cuticle [114]. ColA and ColB have been identified only in coleopteran species, both having bacteriostatic activity against Gram-positive and Gram-negative bacteria. ColA impairs *Escherichia coli* cell division and leads to bacterial gigantism through specific binding with OmpA, which allows the peptide to enter the bacterial cell, where it interacts with a 60 kDa chaperonin GroEL. A similar phenotype was described for *S. pierantonius* cells in the weevil bacteriome [114]. Inhibition of *ColA* expression by RNAi in *Sitophilus* weevils was shown to result in partial loss of endosymbiont control, allowing them to leave the bacteriocytes and infect the surrounding tissues. Moreover, it was found that the *ColA* transcript level in weevils was correlated with the *S. pierantonius* load and that the expression of *ColA* (and *ColB*) in the bacteriome could be modulated by systemic infection of insects by exogenous bacteria [87,115,116].

The bean bug *Riptortus pedestris* (Hemiptera) with its monospecific gut symbionts, i.e., the genus *Burkholderia*, is a useful gut symbiosis model. The midgut of *R. pedestris* is divided into five sections M1, M2, M3, M4B, and M4. In the M4 region, the symbiotic organ contains a specific symbiont *Burkholderia insecticola*, which is orally acquired from the environment by second-instar nymphs [117]. Park et al. showed that the intestinal immunity of *R. pedestris* is related to an AMP called rip-thanatin and that the expression of the *rip-thanatin* gene upon systemic bacterial infection significantly increased in the M4 crypt [118]. When *rip-thanatin* was silenced by RNAi, the number of symbiotic bacteria also increased in the midgut of the bean bug, suggesting that thanatin controls the population of gut-colonizing *Burkholderia* symbionts. Indeed, it was shown that rip-thanatin has high activity against *E. coli* and *Staphylococcus aureus* bacteria but lower activity against symbiotic *Burkholderia.* The *Burkholderia* population has to be controlled in the midgut; when overgrown and present in the hemolymph of host insects, *Burkholderia* exhibit insecticidal activity [117,118,119].

Insect gut microbiota and resistance to infection can also be affected by endosymbionts that reside outside the gut. *Drosophila* flies naturally harbor two such endosymbiotic bacteria: *Spiroplasma* and *Wolbachia* [120,121]. The cell wall-less *Spiroplasma poulsonii* and its relatives are vertically transmitted facultative symbionts residing extracellularly in hemolymph of several *Drosophila* species. The absence of the host immune response to *Spiroplasma* is not caused by suppression of the immune system but is related to the fact that *Spiroplasma* cells are not detected by these flies [122,123,124]. It has been demonstrated that *Spiroplasma* generally did not affect the production of AMPs in *Drosophila* and triggered only mild and chronic synthesis of immune factors. However, proteomic analyses revealed enrichment of attacin and bomanin Bc3 in hemolymph of *Spiroplasma*-harboring insects. Nevertheless, the overexpression of selected AMP genes did not influence the endosymbiont titer, and flies with deleted 10 AMP genes contained *Spiroplasma* titers comparable with the control flies [116]. A comparison of the survival rate of *Spiroplasma*-free and *Spiroplasma*-harboring flies infected with different bacteria and the fungus *Beauveria bassiana* revealed that *Spiroplasma* decreased the resistance of the flies to infection by certain bacterial pathogens, mainly Gram-negative bacteria (*Erwinia carotovora*, *Enterobacter cloacae*) [124].

**Table 1 ijms-24-05753-t001:** Insects and their gut endosymbionts.

Order	Host Insect	Symbionts	Reference
Lepidoptera	*Bombyx mori*	*Proteus vulgaris*, *Erwinia* sp. *Klebsiella pneumoniae* *Citobacter freundii* *Pseudomonas fluorescens*	[125]
	*Galleria mellonella*	*Streptococcus faecalis* (*Enterecoccus mundii*)	[99,126]
	*Plutella xylostella*	*Enterococcus* sp. *Enterobacter* sp., *Serratia* sp.	[127]
	*Spodoptera litura*	*Serratia* sp.	[128]
Diptera	*Aedes aegypti*	*Wolbachia*	[129]
	*Anopheles gambiae*	*Enterobacter asburiae*, *Serratia* sp. *Microbacterium* sp. *Sphingomonas* sp. *Chryseobacterium meningosepticum*	[130]
	*Drosophila melanogaster*	*Spiroplasma poulsonii* *Acetobacter thailandicus* *Lactobacillus plantarum*	[96,131,132,133]
	*Drosophila nebulosa*	*Spiroplasma poulsonii*	[124,134]
	*Glossinia* spp.	*Sodalis glossimidia* *Wigglesworthia* sp. *Wigglesworthia glossinia*	[79,135,136,137,138]
	*Melophagus ovinus*	*Arsenophonus melophagi* *Sodalis melophagi*	[139,140]
Hymenoptera	*Apis mellifera*	*Gilliamella apicola*, *Snodgrassiella* sp. *Frischella perrara*, *Snodgrassella alvi* *Bartonella apis*	[102,103,141,142,143,144,145,146]
	*Bombus* spp.	*Giliamella bombicola* *Snodgrassiella* sp.	[146,147]
	*Camponotus floridanus*	*Blochmannia floridanus*	[112]
Coleoptera	*Acyrthosiphon pisum*	*Buchnera aphidicola*	[148,149,150]
	*Cyrtotrachelus buqueti*	*Lactococcus* sp., *Serratia* sp. *Dysgonomonas* sp., *Enterrococus* sp.	[151]
	*Holotrichia parallela*	*Pseudomonas* sp., *Ochrobacterium* sp. *Cellulosimicrobrium* sp.	[152]
	*Hylobius abietis*	*Erwinia* sp., *Rabnella* sp., *Serratia* sp.	[153,154]
	*Nicrophorus vespilloides*	*Providencia* sp., *Morganella* sp. *Vagococcus* sp., *Proteus* sp. *Koukoulia* sp., *Serratia* sp.	[155,156,157]
	*Pachyrhynchus infernalis*	*Nardonella* sp.	[123]
	*Paradieuches dissimilis*	*Caballeronia*, *Symbiopectobacterium Wolbachia*, *Rickettsiella*	[158]
	*Riptortus pedestris*	*Burkholderia*	[159,160]
	*Sirex noctilio*	*Streptomyces* sp.	[161]
	*Sitophilus oryze*	*Sodalis pierantonius*	[162]
	*Sitophilus weevils*	*Sodalis pierantonius*	[163]
Blattodea	*Blattella germanica*	*Blattabacterium cuenoti*	[164]

The effect of *Wolbachia*, residing mainly in the fly germline, on *D. melanogaster* gut commensal microbiota was reported as well. *Wolbachia* is an intracellular inherited bacterium naturally infecting more than 60% of all insect species worldwide and affecting host populations, e.g., through its positive influence on host fitness [165]. In *Wolbachia*-infected flies, significantly reduced titers of *Acetobacter* species were detected, whereas the titers of other commensal bacteria from the genus *Lactobacillus* were not affected [166]. Interestingly, *Wolbachia* protects *D. melanogaster* against enteric viral infections and those caused by the opportunistic pathogen *Pseudomonas aeruginosa* [167,168]. Therefore, it is regarded as “a supplementary immune system” protecting the host against infection. In turn, other data indicate that *Wolbachia* does not affect the survival and immune response of *D. melanogaster* or *D. simulans* during systemic infection by *P. aeruginosa*, *E. carotovora*, or *S. marcescens* [169] and by intracellular bacteria such as *Listeria monocytogenes* and *Salmonella typhimurium* or extracellular bacterial pathogens *P. rettgeri* [170]. Later, it was found that a key role in the protection of *Drosophila* by *Wolbachia* is played by the route of infection; the protection is ensured in the case of enteric but not systemic infection by *P. aeruginosa.* Gupta et al. observed that the survival of flies after oral infection with *P. aeruginosa* increased when they were *Wolbachia*-positive, in comparison to *Wolbachia*-negative ones. Additionally, it was found that this protective effect was sexually dimorphic, as male flies harboring *Wolbachia* were able to clear bacteria within a week, whereas females stopped infection clearing after 96 h and maintained a stable bacterial load. Expression studies of Imd-mediated genes involved in antibacterial immune response have shown that they are upregulated during enteric bacterial infection in flies. In females, significantly increased expression of *PGRP-LC* in the anterior fly midgut and *PGRP-LE* in the posterior midgut was observed. In turn, increased expression of *AttA*, which is mediated by the Imd pathway, was shown in males [168].

## 6. Non-Classical AMPs and Infection

In addition to peptides exhibiting the characteristics of AMPs, recent research on *Drosophila* has indicated a role of so-called non-classical AMPs in infection. Some of them can confer infection resistance. Interestingly, the expression of these peptides is under the control of the Toll pathway [171]. Taxonomically-restricted *Drosophila*-specific bomanins, a peptide family encoded by 12 *Bom* genes activated by the Toll pathway are an example of non-classical AMPs involved in resistance. It was reported that *Bom*^Δ*55C*^ mutant flies lacking 10 *Bom* genes succumbed to *Enterobacter faecalis*, *Candida glabrata*, and *Fusarium oxysporum* more quickly than wild-type flies due to the defect in resistance [172]. A further study revealed that the individual short-form bomanins can provide resistance against *C. glabrata* when present in *Drosophila* hemolymph in the absence of the remaining 9 Bom peptides. This effect was dependent on the level of transcription rather than on the peptide sequence [173]. Moreover, *Drosophila* mutants in the Bom55C locus exhibited susceptibility to *Aspergillus fumigatus* mycotoxins, verruculogen, and restrictocin. The susceptibility of these mutant flies was rescued by overexpression of bomanins. For example, flies expressing *BomS6* in the nervous system survived better and recovered faster from tremors caused by verruculogen injection, indicating that specific bomanins are able to neutralize mycotoxins instead of acting as classical AMPs [174]. Recently, it has been demonstrated that, from among the *D. melanogaster Baramicin* genes, namely *BaraA*, *BaraB*, and *BaraC*, only *BaraA* is involved in immunity and is upregulated by the Toll signaling pathway upon infection. The peptide products of this gene, i.e., IM10-like peptides, exhibit antifungal activity in vitro [175]. However, results reported by Huang et al. connected the antimicrobial activity of *BaraA*-encoded peptides with neutralization or counteraction of microbial toxins (e.g., *E. faecalis* EntV bacteriocin, *Metarhizium robertsii* destruxin A) rather than with direct antimicrobial action. Interestingly, the two other genes, *BaraB* and *BaraC*, encode truncated forms and are nervous system-specific genes in *Drosophila* [175,176]. *Daisho1* and *Daisho2*, i.e., other taxonomically-restricted genes encoding *Drosophila*-specific non-classical AMPs, have also interesting contribution to antifungal defense. These two genes encode peptides that belong to effectors activated by Toll signaling, likewise *Bom* and *Baramicin* genes. They appeared to be particularly involved in defense against some subset of filamentous fungi, e.g., some species from the genera *Fusarium* and *Aspergillus*. Although in vitro binding of Daisho2 to *F. oxysporum* hyphae was demonstrated, whether Daisho peptides possess direct antifungal activity remains to be elucidated [177].

## 7. AMPs in Antiviral Response

Like all living organisms, insects are exposed to viruses during their lifetime. Studies of interactions between insects and viruses focused initially on species used economically by humans, such as the silkworm *B. mori* or the honeybee *A. mellifera*. A special role in the study of insect immune mechanisms, also in the context of antiviral defense, is played by the fruit fly *D. melanogaster* [178,179].

The primary mechanism involved in the antiviral response in insects is the activation of the RNAi pathway. This pathway is triggered by the presence of viral double-stranded RNA (dsRNA) formed during replication of RNA viruses or, in the case of DNA viruses, formed between complementary transcripts from convergent transcription [180]. In the RNAi pathway, small RNA is produced by cleavage of viral dsRNA. There are three main types of small RNAs: small interfering RNAs (siRNAs), microRNAs (miRNAs), and PIWI interacting RNAs (piRNAs). Damaged viral RNA prevents its replication [179,181]. Activation of RNA-based immunity has been described in a recently published review [179]. In addition, insects engage melanization, encapsulation, endocytosis, and autophagy in response to viral infection [182]. As a result of viral infection, the Toll, Imd, and Jak-STAT signaling pathways are also activated, which leads to induction of the synthesis of active molecules, including AMPs [183].

Representatives of the order Diptera, including many species of mosquitoes, are vectors for human pathogens, e.g., Zika, yellow fever, dengue, and Chikungunya viruses. Viruses in these insects occur in a persistent form, which is associated with the risk of transmission to humans during a bite [184]. Therefore, there are numerous studies on the mechanisms of the antiviral response in insects and the ability to eliminate viruses by the insect’s organism and thus reduce the risk of transmission [185]. Most mosquito viruses are RNA viruses belonging to the family Flaviviridae (yellow fever virus, dengue-DENV, Japanese encephalitis virus, West Nile virus, Zika virus), Togaviridae (Chikungunya virus, Sindbis virus), and Bunyaviridae. Viruses enter the mosquito’s body while feeding, multiply, and then reach its salivary glands. From there, they are transmitted to the new host. The transfer from the midgut to the salivary glands is not fully understood. It is likely that the virus replicates in the midgut epithelium, from where it enters the hemocoel. Then, along with the hemolymph, it can migrate to other tissues, i.e., salivary glands, fat body, trachea, muscles, or neural tissue. The mosquito’s immune system must protect the host by limiting the virus load, which is also a key factor limiting the transmission of the virus to humans [181].

Particular attention is focused on immunological reactions in the mosquito salivary glands, as this is the last stage before the transmission of the virus to humans. Genomic analyses of salivary glands of uninfected *Aedes aegypti* and those infected with DENV showed that the infection causes an increase in the expression of genes activated by the Toll and Imd pathways, including the gene encoding cecropin-like peptide. Both the pre-mature and mature forms of the peptide showed antiviral activity against the DENV and Chikungunya virus [186]. AMPs were also found in the *Ae. aegypti* midgut, fat body, and hemocytes. Activation of the Toll and Jak-STAT pathways in *Ae. aegypti* has been shown to reduce the level of DENV replication in the midgut [187]. Pan et al. confirmed the antiviral effect of AMPs in *Ae. aegypti* against DENV. Their research showed that defensins and cecropins made the *Wolbachia*-infected mosquito resistant to this virus. Mutations in the genes encoding defensin C, defensin D, cecropin N, and lysozyme C resulted in a higher viral load in the mosquito’s body. There was no antiviral activity of defensin A, cecropin G, cecropin D, cecropin E, and gambicin [185,188,189,190].

Activation of the Toll pathway in *D. melanogaster* is triggered after infection with *Drosophila* X virus (DXV), *Drosophila* C virus (DCV), Cricket paralysis virus (CrPV), Flock House virus (FHV), and Nora virus. Research conducted by Ferreira et al. indicated that the Toll pathway determined the immunity of *Drosophila* after natural oral infection with DCV, CrPV, FHV, and Nora virus. Toll pathway mutants showed a higher viral load [191]. It has also been shown that *Drosophila Dif* mutants were more sensitive to DXV [192]. This was not found for *Dif* or *Dif/Dorsal* mutants against DCV or Sindbis virus (SINV) [193,194]. In bees, knocking out the Dorsal-encoding gene increased the load of DWV [195].

In their research on *D. melanogaster* infected with DNA Kallithea virus, Palmer et al. indicated an evolutionarily conserved role of the Toll pathway in the antiviral response to DNA viruses. They showed that RNAi mutants and *Imd* mutants were susceptible to infection with this virus. Interestingly, no such observations were reported for *Toll* mutants. The researchers identified the Kallithea virus gp83 protein, which inhibits the Toll pathway by regulating the activity of NF-κB transcription factors. A similar effect was found for the *Drosophila innubila* nudivirus (DiNV) gp83 protein [196].

Huang et al. showed that infection of *Drosophila* with SINV activated the Imd and Jak-STAT pathways (Figure 1). The activation of the Imd pathway resulted in an increased number of transcripts of genes encoding metchnikowin (*Mtk*), diptericin B (*DptB*), and attacin C (*AttC*). This study also indicated that *Drosophila* with the knockdown of *AttC* and *DptB* genes showed a higher level of SINV virus [184]. The exact mechanism of the antiviral action of these AMPs is not fully known. It is hypothesized that attC may inhibit viral replication by interacting with viral or cellular components of the replication complex. In contrast, dptB does not affect the viral replication process; however, the synthesis of this peptide is associated with a lower viral load in *Drosophila* infected with SINV. In addition, an important antiviral role of dptB was found during the insect development. A mutation in the *dptB*-encoding gene resulted in abnormal development of flies when accompanied by SINV infection [184]. The Jak-STAT pathway was activated in response to infections with DENV, DCV, West Nile virus, and SINV. In both mosquitoes and *Drosophila*, inactivation of Jak kinase resulted in the development of viral infection [184]. It is hypothesized that this pathway is activated in *Drosophila* in response to cell damage due to viral infection rather than in response to viral signaling molecules [179].

The interesting results published by Zhang et al. indicated a non-canonical pathway leading to the synthesis of AMPs in response to viral infection. As mentioned earlier, RNAi is the main antiviral defense mechanism in insects. In order to counteract this mechanism, viruses produce viral suppressors of RNAi (VSRs), which can target all steps of the RNAi pathway. The authors described *Drosophila* VSR-interacting long non-coding RNA (VINR), which is involved in the induction of AMP gene expression upon detection of dsRNA-binding VSR of *Drosophila* C virus via a non-canonical Cactin-Deaf1 pathway [197].

AMPs are also involved in the antiviral response of *B. mori* [198]. This insect species is exposed to attack of e.g., *B. mori* nucleopolyhedrovirus (BmNPV), *B. mori* densovirus type 1 (BmDNV-1), and *B. mori* bidesovirus (BmBDV) also referred to as *B. mori* densovirus type 2 (BmDNV-2). BmNPV infection of both susceptible and resistant insects was found to result in the induction of AMPs, i.e., gloverin-1, -2, -3, lebocin, cecropin, attacin, and lysozyme. Since the increased expression of the gene encoding gloverin-4 was detected in resistant insects only, this peptide was recognized as an AMP conferring *B. mori* resistance to BmNPV [198,199]. In *Helicoverpa armigera* infected with the baculovirus *Autographa californica* multiple nucleocapsid nucleopolyhedrovirus (AcMNPV), the expression of five genes, including those encoding gloverin, cecropin-1, galiomycin, and lysozyme, was increased [200]. Moreno-Habel et al. used two gloverins from *Trichoplusia ni* hemocytes infected with AcMNPV. They pointed to the probable mechanism of the antiviral activity of gloverins, which may cause destruction of the BV envelope by accumulating on the surface of BVs [201]. Although *G. mellonella* (an apiary pest) is a commonly used model organism to study bacterial and fungal pathogens, there are insufficient literature data on the immune response of *G. mellonella* to viral infections. Traiyasut et al. showed that the total RNA of both *Apis cerana japonica* worker bees and *G. mellonella* contained nucleotide sequences of Israeli Acute Paralysis Virus (IAPV) and Black Queen Cell Virus (BQCV) [202]. Software used to predict the antiviral activity of AMPs indicated that some *G. mellonella* AMPs may have such activity [203]. The results obtained by us using Meta-iAVP (available at http://codes.bio/meta-iavp/; accessed on 9 December 2022) are presented in Table 2.

Like other insects, bees and bumblebees are not defenseless against viral pathogens. This is of particular economic importance from the human point of view due to the invaluable role of these representatives of the Hymenoptera order as pollinators. Most of the viruses that attack *A. mellifera* are RNA viruses, such as Dicistroviruses (BQCV, IAPV) and Iflaviruses (DWV, sacbrood virus, slow bee paralysis virus). The antiviral response involves similar mechanisms to those in Diptera and Lepidoptera. The activation of the Toll and Imd signaling pathways leads to production of many AMPs, including Hymenoptera-specific hymenoptaecin, abaecin, and apidaecin as well as other AMPs like defensin or lysozyme [182]. In the bumblebee *Bombus terrestris*, the level of hymenoptaecin gene expression increased after injection of IAPV [204]. Similarly, *A. mellifera* infection with DWV as well as concomitant exposure to DWV type A and a pesticide thiamethoxam resulted in increased expression of the hymenoptaecin gene [205,206]. Mookhploy et al. showed that the level of hymenoptaecin and defensin expression increased after injection of DWV into *A. mellifera* pupae, while abaecin and hymenoptaecin genes were upregulated in newly emerged adult bees [207].

## 8. AMPs in Insect Venoms

Among insects, venom is produced by representatives of the order Hymenoptera. Insect venom secreted by the venom glands is a rich source of bioactive molecules, including proteins, peptides, nucleotides, free amino acids, amines, and inorganic salts. In addition to large proteins with enzymatic activity, e.g., phospholipases, there are also peptides with a mass below 10 kDa [208]. The venom composition depends on its function. In social hymenopteran species, the venom is used both to defend the insect against predators and to incapacitate the prey. In solitary and parasitic wasps, the venom additionally contains neurotoxic substances. It also has antimicrobial functions preventing the spread of infection, the source of which may be the prey brought to the nest. The antimicrobial functions are connected e.g., with the presence of AMPs in the venom. As other peptides with antimicrobial activity, such peptides are usually amphipathic molecules.

The most common venom-producing animals are ants belonging to the family Formicidae [209]. Because ants live in high-density colonies, they are at high risk of infection and spread of pathogens. A potential source of infection in predatory species can be food; therefore, in addition to cytolytic peptides, potent antimicrobial molecules must be present in the venom. Of major importance are linear cationic peptides with antibacterial activity. Particularly rich in AMPs are the venoms of ants from the subfamilies Ponerinae, Paraponerinae, Myrmicinae, Myrmeciinae, Pseudomyrmecinae, Ectatomminae. The tropical ponerine ants *Neoponera goeldii* (previously *Pachycondyla goeldii*) produce amphipathic α-helical ponericins [210]. They have bactericidal effects against Gram-positive and Gram-negative bacteria as well as cytolytic and insecticidal activity. They are classified into three main families, depending on the sequence homology to previously characterized AMPs, e.g., ponericin G shows similarity to insect cecropin-like peptides. Ponericin W shows homology to bee melittin and frog gaegurin, while ponericin L is similar to dermaseptin of *Phasmahyla* and *Phyllomedusa* frogs [211,212,213]. The Australian ant *Myrmecia pilosula* is responsible for 90% of life-threatening allergic reactions to ant stings. Two main protein allergens from the venom of these ants have been identified: pilosulin 1, which exhibits cytolytic and hemolytic activity, and pilosulin 2 (pilosulin 3a), which occurs in the venom only in the form of a heterodimer (with pilosulin 3b) called pilosulin 3. It constitutes 80% of the peptides contained in venom and exhibits antibacterial and antifungal activity [214,215]. The venom of ants living in South America, *Dinoponera quadriceps* and *Dinoponera australis*, contains ponericin- and pilosulin-like peptides collectively called dinoponeratoxins, which are classified into six groups. Group III dinoponeratoxins show homology to AMPs from the temporin family, originally isolated from skin secretions of frog *Rana temporaria*. The high homology in the distribution of proline and leucine residues in both peptides suggests that these dinoponeratoxins exhibit antimicrobial activity similar to that of frog temporins. Temporins increase microbial membrane permeability and influence intracellular metabolic processes, probably without affecting cell integrity. Group V contains antibacterial and antifungal peptides isolated from *D. quadriceps*, which are similar to ponericin W, dinoponeratoxins of *D. australis*, and poneratoxins. Multifunctional dinoponeratoxins with antimicrobial, hemolytic, and histamine-releasing properties have been characterized. They were demonstrated to be active against bacteria, fungi, and parasites [209,216,217,218]. The dominant peptide in *A. mellifera* venom is melittin, constituting 40–60% of its dry weight [219]. Melittin is composed of 26 amino acids, which are spatially arranged in an α-helical structure. With an increasing peptide concentration and pH value in the environment, melittin monomers form tetrameric structures. Due to its amphipathic character, melittin interacts with cell membranes, causing their disintegration. It can act synergistically with phospholipases, facilitating hydrolysis of fatty acids. In addition to its broad antibacterial and antifungal activity, melittin exhibits strong hemolytic action [220]. Apamin and mast cell degranulating peptide (MCDP) have also been found in honeybee venom. Apamin and MCDP consist of 18 and 22 amino acids, respectively. The structure of both peptides is stabilized by two disulfide bridges. Apamin and MCDP have a neurotoxic effect by blocking Ca^2+^-dependent potassium channels or binding to specific receptors [221,222,223]. Secapins are other peptides found in the venom of the honeybees: secapin-1 from European and Chinese bees and secapin-2 from African bees. These 25 amino acid peptides contain one disulfide bridge and function as serine protease inhibitor-like peptides. Interestingly, *A. cerrana* secapin-1 was demonstrated to have antimicrobial activity. The peptide can bind to bacterial and fungal cell surfaces [224,225]. Melectin is an AMP isolated from the venom of the cleptoparasitic bee *Melecta albifrons*. This peptide consists of 18 amino acids and forms an α-helical structure. It can bind to lipopolysaccharide or lipoteichoic acid, leading to rapid death of a broad spectrum of Gram-negative and Gram-positive bacteria through membrane permeabilization. In contrast to melittin, melectin exhibits low hemolytic activity [226,227].

Bombolitins are functional equivalents of melittin in the venom of bumblebees of the genus *Bombus*. They share structural and biological properties with melittin. Depending on the species, they contain 17–19 amino acid residues [228]. In general, their interaction with synthetic membranes induces formation of α-helical structures integrated into lipid bilayers; however, differences in antimicrobial activity between bombolitins from different species have been demonstrated [229]. In addition to antimicrobial activity, they have the ability to lyse erythrocytes and liposomes, increase the activity of phospholipases, and release histamine from mast cells [210].

In the venom of various species of wasps and hornets from the Vespidae family, mastoparans capable of degranulating mast cells are functional analogs of melittin and bombolitin. Mastoparans typically contain 14 amino acids with high content of hydrophobic and basic residues; they form amphipathic α-helical structures. Similar peptides were found in the venoms of social wasps of the genus *Polistes*: mastoparan in *Polistes jadwiga* and dominulin A and B in *P. dominulus*. These peptides exhibit activity against Gram-positive and Gram-negative bacteria based on the formation of pores in the cell membrane [230]. Another example of venom peptides with antimicrobial activity is the 13 amino acid long crabrolin, originally described in the venom of the European hornet *Vespa crabro* and classified as a chemotactic factor. Its activity against selected bacteria is determined by the presence of hydrophobic groups and the positive charge, while the α-helical conformation is necessary to maintain hemolytic activity [231,232].

## 9. Conclusions

The involvement of AMPs in insect immune response against invading pathogens has long been recognized as a primary role of these molecules in the insect body. AMPs, synthesized particularly in the fat body and secreted into hemolymph, are essential components of systemic immune response to microbial infections. However, due to the specific properties and proper regulation of their gene expression, AMPs not only are able to fight pathogenic infections but also can control symbiotic microflora in the insect gut. Their strong antimicrobial properties make AMPs essential components of both royal jelly and insect venoms, protecting insect colonies against spread of infections. The ability of AMPs to interact with phospholipid membranes facilitates their binding to altered membranes of some cancer cells, which results in anticancer activity in vivo. Although the modes of the antiviral action of insect AMPs are still not well understood, the increased expression of particular AMP genes in response to viral infection clearly indicates their role in fighting viral pathogens. In addition to their purely antimicrobial action, AMPs are multifunctional molecules that highly contribute to proper functioning of the insect brain and the whole nervous system and to maintenance of intestinal homeostasis by preventing microbiota dysbiosis. Dysregulation of AMP expression may result in shortening of the insect lifespan and contributes to aging. AMPs are important players in the cross-talk between innate immunity, nervous system activity, microbiota homeostasis, and metabolism in insects.

## Figures and Tables

**Figure 1 ijms-24-05753-f001:**
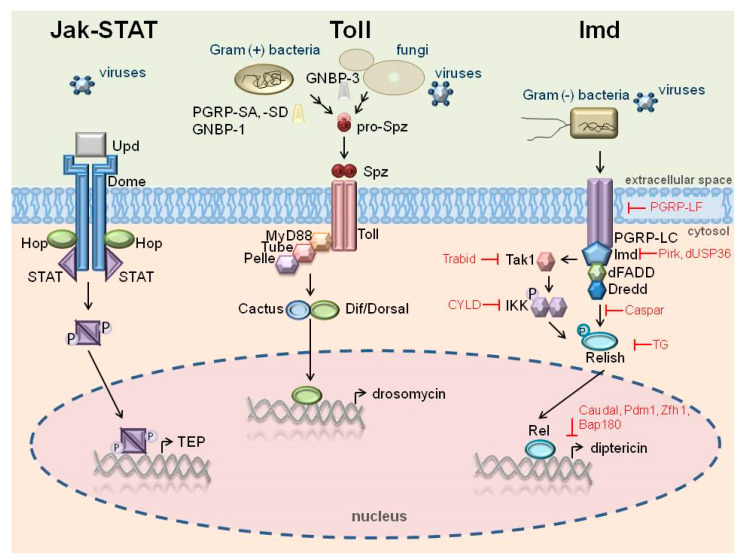
Main signaling pathways involved in AMP gene regulation in response to infection (simplified scheme). The **Jak-STAT** (Janus kinase/signal transducers and activators of transcription) pathway is induced by septic injury/damage signals or bacterial or viral infection through binding of Upd (Unpaired) cytokine to the receptor Dome (Domeless). Next, Hop (Jak kinase Hopscotch) phosphorylates Dome and STAT (Stat92E) bound to the receptor. STAT disconnects, dimerizes, and translocates into the nucleus, where it induces gene expression, e.g., TEP. The **Toll** pathway is activated by binding the ligand Spz (Spaetzle), which is formed from pro-Spaetzle by SPE (Spaetzle-processing enzyme) cleavage when extracellular recognition receptors PGRP-SA, -SD, and GNBP-1 recognize Gram-positive bacteria (Lys-type PGN) or GNBP-3 recognizes fungi (β-1,3-glucan), as well as after recognizing viral infection. The activated Toll receptor (dimer) binds the adaptor protein MyD88, which recruits Tube and the kinase Pelle to form a MyD88-Tube-Pelle complex. Pelle phosphorylates Cactus (IκB) leading to its degradation. Dorsal/Dif transcription factors released from the complex with Cactus translocate into the nucleus, where they activate the expression of AMPs. The **Imd** pathway is activated by transmembrane PGRP-LC receptors after recognition of Gram-negative bacteria (DAP-type PG). An adaptor protein Imd interacts with dFADD (*Drosophila* Fas-associated death domain) and Dredd (Death related ced-3/Nedd2-like caspase). Activated Tak1 kinase (TGF-beta activated kinase 1) activates IKK. The NFκB factor Relish is phosphorylated by active IKK and cleaved by Dredd to release the Rel-68 domain, which translocates into the nucleus to activate transcription of target genes. Negative regulators of the Imd pathway are indicated by 
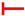
. The Imd pathway can also be activated by viruses.

**Figure 2 ijms-24-05753-f002:**
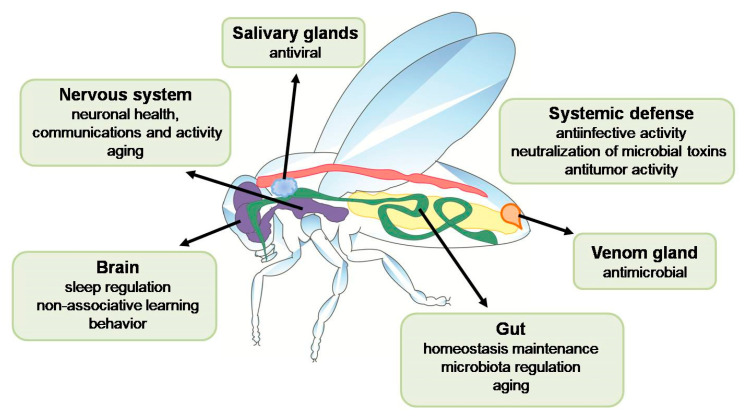
Different functions of AMPs in insects.

**Table 2 ijms-24-05753-t002:** Predicted antiviral activity of selected *G. mellonella* AMPs.

*G. mellonella* AMP	Accession Number/ Peptide ID	Predicted Antiviral Activity	AVP Value	Non-AVP Value
gallerimycin	XP_026765221.1	+	1	0
lebocin-like anionic peptide 1	P85211.1	+	0.98	0.02
cecropin D-like peptide	P85210.1	+	0.93	0.07
*G. mellonella* cecropin	XP_026754247.1	+	0.746	0.254
apolipophoricin	P80703	+	0.698	0.302
proline rich peptide 2	P85212.1	−	0.324	0.676
defensin-like peptide	P85215	−	0.272	0.728
defensin	P85213	−	0.268	0.732
proline rich peptide 1	P85214.1	−	0.004	0.996
anionic peptide 2	P85016	−	0	1

The amino acid sequences and accession numbers of *G. mellonella* AMPs were obtained using the NCBI Reference Sequences database. The antiviral potential (AVP) was predicted using Meta-iAVP available at http://codes.bio/meta-iavp/; accessed on 9 December 2022).

## Data Availability

Not applicable.

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
