# Peer review of "Unraveling the Role of Antimicrobial Peptides in Insects"

_ijms, 2023, doi:10.3390/ijms24065753_

Round 1
Reviewer 1 Report
ijms-2271528 Unraveling the role of antimicrobial peptides in insectsStączek and co-authors desribed the role of antimicrobial peptides (AMPs) in insect. Most literature data is focusing of the role of AMPs as an antibacterial, antifungal, and antiparasitic factors, while presented manuscript described the not obvious functions of AMPs in the insect. Authors pointed the antiviral activity of AMPs and their role in the regulation of brain-controlled processes e.g. sleep and nonassociative learning and functioning of the insect nervous system, by influencing neuronal health, communication and activity. They reviewed the strong antimicrobial properties of AMPs, which are essential components of both royal jelly and insect venoms, protecting insect colonies against spread of infections. Authors described the potential anticancer activity in vivo of AMPs, based on their ability to interact with phospholipid membranes, which facilitates their binding to altered membranes of some cancer cells. Article is well-written and is a significant contribution to the field. The work is well organized and comprehensively described and the appropriate and adequate references to related and previous work are used.
Author Response
The authors thank the Reviewer for the opinion.
Reviewer 2 Report
This is a very timely review, in which Authors describe the state of art concerning antimicrobial peptides in insects, enlightening their novel and multiples roles beyond the immune one. I firmly believe that this review will be of broad interest to the readers.
Only one suggestion: add and discuss the reference Barati et al. concerning AMP upregulation in a Drosophila model of Alzheimer disease.
Author Response
The authors thank the Reviewer for all comments and suggestions which allowed us to improve the manuscript.
In the revised manuscript, information based on the suggested literature has been added in the section AMPs in the nervous system, lines 205-214, as follows “In turn, Barati et al. studied the effect of amyloid-b 42 (Ab42) and tau on the Imd pathway and neuroinflammation gene expression in a Drosophila model. They showed that the expression of genes involved in the Imd pathway, such as Relish and AMPs (AttA, DptB), increased with age in the W1118 control flies and in the embryonic nervous system of AD transgenic flies Ab42, tauWT, or tauR406W, but the level of AMPs in glia remarkably decreased compared to W1118. The decline was higher in both tau flies compared to A42 transgenic flies. The overexpression of AMPs in Drosophila leads to brain neurodegeneration and neuroinflammation [42]”, and papers cited in the References have been added.
Barati, A.; Masoudi, R.; Yousefi, R.; Mosefi, M.; Mirshafiey, A. Tau and amyloid beta differentially affect the innate immune genes expression in Drosophila models of Alzheimer’s disease and b-D monnuroic acid (M2000) modulates the dysregulation. Gene 2022, 808, 145972-145984.
Reviewer 3 Report
A well written and comprehensive review. This paper is a valuable resource as it provides an overall look at the AMP with their role and functions.
As a suggestion for the authors I would recommend them to include some more detailed information about the structural characteristics of AMP; both generally, but also if possible, to make some remarks considering structure-activity relationships. E.g. do AMP with a common role have a common structural motif.
Author Response
First of all, we would like to thank the Reviewer for the suggestion, which was very helpful and enabled us to improve the manuscript. Please, find below our response to the comment.
Since our goal was to present the less-known role of AMP, but not the antimicrobial activity of peptide, the related structure of molecules, their division, and mechanisms of action, this description was omitted in our work. Especially since there are now many interesting reviews on this subject.
In the revised manuscript, the suggested information has been briefly added in the Introduction, lines 51-67, as follows “Insect AMPs can be divided into three classes: (i) linear peptides without cysteines forming α-helices, (ii) peptides containing cysteines and stabilized by disulfide bridges, e.g. peptides with β-sheet structures or αβ motifs, and (iii) peptides with overrepresentation of one amino acid, usually glycine and/or proline. Despite the broad structural diversity, studies on the structure-activity relationships revealed that (i) net charge, (ii) hydrophobicity, and (iii) amphipathicity are the most important physicochemical and structural determinants of antimicrobial activity and effectiveness as well as selective toxicity of AMPs [6,8,13]. Moreover, using bioinformatic and pattern recognition methods, multidimensional structural signatures were identified in AMPs, which appeared to be the requisites for antimicrobial activity. One of such unifying signatures was found in eukaryotic α-helical AMPs. This unifying α-core signature contains a helical domain of 12 residues with a mean hydrophobic moment of 0.50 and favoring aliphatic over aromatic hydrophobic residues [14]. Similarly, a conserved signature, called the γ-core motif, was identified in AMPs stabilized with disulfide bridges. The γ-core motif is composed of two antiparallel β-sheets with a short turn region. The sequence signatures include (i) the length of 8–16 amino acid residues, and (ii) conserved GXC or CXG motifs within the sequence [15].”, and papers cited in the References have been added.
Yount, N.Y.; Weaver, D.C.; Lee, E.Y.; Lee, M.W.; Wang, H.; Chan, L.C.; Wong, G.C.L.; Yeaman, M.R. Unifying structural signature of eukaryotic α-helical host defense peptides. Proc. Nat.l Acad. Sci. U.S.A. 2019, 116, 6944-6953.
Yount, N.Y.; Yeaman, M.R. Multidimensional signatures in antimicrobial peptides. Proc. Natl. Acad. Sci. U.S.A. 2004, 101, 7363-7368.
Hafeez, A.B.; Jiang, X.; Bergen, P.J.; Zhu, Y. Antimicrobial peptides: an update on classifications and databases. Int. J. Mol. Sci. 2021, 22, 11691-11743.